# What Do You Think of AI? Research on the Influence of AI News Anchor Image on Watching Intention

**DOI:** 10.3390/bs12110465

**Published:** 2022-11-21

**Authors:** Ke Xue, Yifei Li, Hanqing Jin

**Affiliations:** School of Media and Communication, Shanghai Jiao Tong University, Shanghai 200240, China

**Keywords:** AI anchors, news anchors, watch intention, perceived attractiveness, inherent cognition

## Abstract

Since the concept of artificial intelligence was introduced in 1956, AI technology has been gradually applied in various fields, including journalism. This paper focuses on research related to AI news anchors, and two correlated experiments are applied to examine audiences’ perceived attractiveness of AI news anchors from a psychological dimension. Study 1 focuses on the different variables that influence the behavioral willingness of AI news anchor viewers, while Study 2 focuses on the mediating and moderating variables that influence audiences’ psychological changes. The results indicate that non-humanoid female AI news anchors who use anthropomorphic voices to broadcast news obtain the highest perceived attractiveness among audiences. Additionally, the mediating effect of perceived attractiveness and the negative moderating effect on the inherent impression of traditional news anchors are both verified in the study. Based on the research findings, the implications and suggestions are addressed accordingly.

## 1. Introduction

As early as 1956, John McCarthy proposed using machines to simulate human intelligence and introduced the concept of artificial intelligence (AI) at the Dartmouth Conference. Since then, AI has emerged as an independent subject. It has begun to develop in a multi-disciplinary direction and has gradually entered various fields of society, including the news industry today. AI technology has been applied in the news field in several ways, such as robot writing, big data algorithm recommendation systems, virtual reality (VR) and augmented reality (AR) applications, and AI news anchors [1]. In 2007, the American StatSheet company introduced automated news. In 2015, “Toutiao” used AI in the collection and distribution of news and employed algorithms to collect users’ data footprints, which were utilized to analyze their reading preferences and ultimately make accurate personalized recommendations. On 7 November 2018, at the Fifth World Internet Conference, the Xinhua News Agency and Sogou jointly released the world’s first synthetic news anchor using the most advanced AI technology to create a news broadcast with the same capabilities as live anchors. AI news anchors have led to technological innovations and breakthroughs in the global AI synthesis field, setting a precedent for real-time audio and image synthesis in the news industry [2].

The research in this field began with the history of the development of AI news anchors and their applications in the industry [3]. As the research continues, there are different controversies in the academic community regarding the adoption of AI technology in the field of journalism. One of the parties pointed out that due to the solidification of the news broadcast model and the continuous development of integrated technology, live anchors can easily be replaced by robots [4], cutting production and employment costs in half and enabling news broadcasts to be online in real-time throughout the day [5]. However, the change led by artificial intelligence has also raised concerns in the journalism industry that the problems of opacity, bias, and discrimination associated with AI technology will be magnified as it begins to be applied in the production and broadcasting of news [6]. Some scholars have even raised the concern that humanoid news will be marginalized and ultimately replaced by AI [7]. From a technological development point of view, Sogou CTO Chen Wei mentioned that the first AI news anchor is still in the first stage of AI development, and it will take a long time to reach the second and third stages, namely, artificial general intelligence (AGI) and artificial super intelligence (ASI). Although AI synthesis technology has achieved a certain degree of breakthrough, it is still in its infancy, and it will take time before it can truly achieve the broadcasting effect of a real person.

Indeed, AI news anchors are one of the main aspects that have visibly tried to blur the boundaries between humans and technology and between what is considered human-like and machine-like matter. Thus, they prompt new inquiries into the communication effects of different styles of news anchors. Many studies have analyzed the advantages and disadvantages of live-action anchors and virtual anchors, mainly focusing on the evaluation of voice expression, appearance, artistic aesthetics, and emotional analysis [8]. The academic community generally believes that current AI news anchors can only produce and broadcast simple, mechanized, and highly repetitive information and have not yet reached the ability to engage in human creative activities. Even though the current AI technology has not been able to participate more deeply in the creative production of news content, the public’s emotions toward AI news anchors are still mainly positive [9]. The conflict between the high acceptance of AI news anchors and the lack of an effective application strategy becomes increasingly severe [10]. However, whether different types of AI news anchors have an impact on the audiences’ perceptions and the mechanisms of these impacts remain to be explored, and research in this area is still in its nascent stage. Only a few scholars have conducted research on AI news anchors from a quantitative perspective. The lack of vivid data and results makes the production of AI news anchors even harder in the journalism industry.

Therefore, this study aims to explore the influence of different AI news anchors on the audiences’ perceptions of the attractiveness of such anchors, and it then explores the public’s willingness to watch news videos hosted by these anchors. Through exploratory research on the factors that influence the public’s willingness to watch AI news anchors, this article attempts to find the most influential factors and the intermediary variables that have an impact on their effects. In addition, based on existing qualitative studies, we identify the influencing factors of the public’s intention to watch AI news anchors through a two-step experimental method to lay the foundations for further detailed research by scholars. On the basis of the public’s inherent impression of news anchors, this paper attempts to explore whether this common stereotype will have the same moderating effect on AI news anchors.

In 2005, Clifford Nass pointed out that people are more receptive to a “female voice” than a “male voice” to assist them in solving problems [11]. This research led to the use of a female voice as the default setting for Siri since its inception. However, no significant academic research has been conducted in the field of AI news anchors to drive the development of the industry, and, thence, this paper is expected to provide a quantitative research conclusion that can provide practical guidance for creating a more popular AI news anchor image and bridge the gap between what the public expects from AI news anchors and what the industry creates.

## 2. Theory and Literature Review

### 2.1. Research on the Influencing Factors of AI News Anchor Images

Existing studies have shown that the attractiveness of external images is an important factor in the effectiveness of communication and an indispensable dimension for individuals to evaluate others [12]. Baker Churchill [13] proposed that attractiveness depends on familiarity and similarity. Based on Churchill’s theory, Huang Yao [14] explored the influence of uniqueness, attractiveness, and brand attitude on virtual images, and concluded that the external image attractiveness of virtual images is still very important in the digital age and that the uniqueness of the appearance can win the public’s favor easier. For the appearance of virtual characters, some scholars have proposed that virtual characters can simulate the appearance of real people and digitally customize hairstyles, clothing, and makeup to suit different occasions, which is a technological advantage [15]; however, other scholars oppose it. According to the uncanny valley effect (1969), when a robot resembles a real person to a certain extent, people will experience subtle psychological discomfort. Lou [16] pointed out that although virtual characters can be very similar to real people and have natural expressions with the support of current technology, audiences can still distinguish the difference from real people, which makes people feel “discomfort” or “even fear” and reduces the attractiveness of virtual characters. Since an AI news anchor is also an extension of a virtual character, this paper considers the appearance of a news anchor to be important and explores whether the appearance of different AI news anchors will affect the audience’s perception of their attractiveness. Therefore, based on the literature review, this paper proposes the following research hypothesis:

**H1.** 
*Compared with humanoid AI news anchors, the image of non-humanoid AI news anchors can positively influence the audience’s perception of the attractiveness of AI news anchors.*


In addition to appearance, research has found that anchors of different genders have different effects on audiences’ emotions, including during clothing brand live broadcasts [17] and other fields. Gender differences, including the impact of gender differences on live anchors and virtual images, have become popular in image research in recent years [18,19]. The same conclusion was also confirmed by Woojung [20], who found that an audience’s perception of the credibility of female anchors is more positive than that of male anchors. Most existing studies have found that compared with male robots, female robots are often considered to have characteristics related to warm interpersonal relationships [21], and people are also more willing to let female robots participate in work involving interpersonal interaction, while male robots are more often used in mechanical and computational work. However, whether these conclusions are still effective when it comes to AI news anchors remains unknown. There is an urge for us to verify whether the gender of virtual characters still plays a stronger positive role. Therefore, this paper proposes the following assumption:

**H2.** 
*Compared with male AI news anchors, female anchors can positively influence an audience’s perception of the attractiveness of AI anchors.*


Exciting research suggests that a sweet sound compared to a harsh sound can lead to a more pleasant and positive experience for the audience [22]. Therefore, voice has become one of the core factors for all language workers, and news anchors are no exception. Researchers have found that creating an ideal sound brand helps to build a TV host’s personal image, which largely affects the audiences’ perceptions of the host’s attractiveness [23]. Having an affinity for voice, mastering the correct oral expression skills, controlling a breathing rhythm, and being able to pronounce correctly, among other things, can allow viewers to better understand news content and, at the same time, enhance the attractiveness of news programs and the viewer’s willingness to watch [24]. Psychological experiments have proved that if an AI virtual image can be visually and auditorily synchronized with real people in real time, it can influence the audience to establish and strengthen the positive emotions of the virtual image [15]. This phenomenon is called the “result effect” and is also known as the “self-effect”, which states that when people classify others as being the same type as themselves, they are more likely to trust those “others” and accept them more easily [19]. Therefore, this paper proposes the following assumption:

**H3.** 
*Compared with non-anthropomorphic voices, anthropomorphic voices can positively affect an audience’s perception of the attractiveness of AI news anchors.*


### 2.2. Research on the Attraction of AI News Anchors

Existing studies have shown that visual attractiveness is an important factor that affects an audience’s attitude and behavioral willingness, and building a positive and attractive image in a person’s mind can effectively trigger their consumption intentions [25,26]. At the same time, image attractiveness can also significantly affect purchase behavior [26,27,28]. In the field of AI, users may anticipate what they will see and what will happen based on their own understanding and cognition of AI. Users’ expectations of AI interactions are based on the mental model of how they perceive AI, and this is the driving factor for their satisfaction and willingness to use AI information [29]. Although there are few pieces of literature on the influence of AI news anchors’ attractiveness on watching intention, based on previous studies, it is not difficult to find a strong correlation between “perceived attractiveness” and “behavioral intention”. Therefore, this paper proposes the following assumption:

**H4.** 
*An audience’s perceived attractiveness to AI news anchors can positively affect the audience’s willingness to watch news videos hosted by AI news anchors.*


### 2.3. Research on Audiences’ Inherent Impression of News Anchors

A “news anchor” has become a normal mode of communication as a news reporter and presenter, and it has become a well-known professional term inside and outside the industry, as well as a common identity mark on TV screens [30], which has almost carved an indelible inherent impression in the mind of the public. Based on previous studies, it can be found that “better language ability and image quality” [31], “plain and dignified appearance”, “profound humanistic accomplishment and emotional expression” [24], “appropriate and sophisticated language expression” [32], and other characteristics have become the expectations of a “news anchor”. Research on AI news anchors has pointed out that the current AI news anchors still have some disadvantages, such as the inability to express and convey emotions, the lack of eye contact, a news broadcasting tone that is too flat and lacks fluctuation [33], and the inability to conduct news. Emotional distinction, identification, and interpretation [34], among other issues, have formed a conflict with the audience’s inherent perception of traditional news anchors. We are concerned that such conflict may contribute to a negative influence on people’s perception of AI news anchors. Therefore, this paper proposes the following assumption:

**H5.** 
*A viewer’s inherent impression of news anchors will negatively regulate the influence of AI news anchors’ perceived attractiveness on the viewer’s willingness to watch news hosted by AI.*


### 2.4. Research on the Watching Willingness of AI News Anchors

With the continuous development of AI technology, AI smart products are gradually becoming more popular and entering the lives of the public. Existing research shows that the audience’s willingness to use AI products and applications, including smart medical care, AI advertising, and AI news, is affected by many external factors [35]. A study by Li and Li Yang [36] found that there is a correlation between AI anchors and the audiences’ willingness to watch. The mimicry appearance and voice interaction of AI virtual anchors positively affect the audience’s willingness to watch. Additionally, Xu [37] found out through related research that the appearance and technological characteristics of AI news anchors have an impact on the audiences’ willingness to watch and challenged traditional TV news anchors. In the field of webcasting, the audience’s viewing intention and behavior are also affected by the external environment and their own experiences [38,39]. The viewing, participation, and consumption behaviors of viewers are affected by the novelty of live content, users’ internal psychological states, and anchor characteristics [40,41,42]. Research shows that the willingness to watch, as an important behavioral willingness for the public to accept a new thing, has received much attention from scholars. Therefore, this paper takes the willingness to watch news from AI news anchors as the dependent variable to explore the influencing factors of the public’s willingness to watch news from AI news anchors.

**H6.** 
*Perceiving the attractiveness of AI news anchors plays an intermediary role in the influence of AI news anchors’ image on viewing intentions.*


## 3. Overview of the Studies

The paper employed a 2 (appearance: humanoid or non-humanoid) × 2 (gender: male or female) × 2 (voice: anthropomorphic or non-anthropomorphic) between-subject design to investigate the users’ willingness to watch AI news. Study 1 focuses on the different variables of AI news anchors and shows the main effects that non-humanoid female AI news anchors with anthropomorphic voices lead to a greater willingness to watch. Study 2 uses an analysis of mediation and moderation to corroborate the role of perceived attractiveness and the inherent impression of news anchors in explaining the processing and evaluating systems while watching different AI news anchors. Study 2 shows different perceptions that result from the different inherent impressions; the powerful inherent impression of traditional news anchors leads to a lower evaluation of AI news anchors, whereas perceived attractiveness plays an intermediary role in the influence of AI news anchors’ images on viewing intentions.

## 4. Study 1

Study 1 aimed to verify whether the different images of AI news anchors would lead to a different degree of willingness to watch AI news anchors. The variables included “appearance”, “gender”, and “voice,” which were tested by three different influencing factors on the audience’s willingness to watch. A single news article and eight different AI news anchors were used as stimuli. The news article, which was used as the research material, was adapted from Xinhua News, which is a national news media agency. This news article consists of 159 words, and the news video clip is 35 s long. The news content briefly introduces the time, place, theme, and composition of the participants of the Global World Summit of Young Scientists. This article was selected because the topic was neutral for the participants. To manipulate the news anchor image, four different news anchor videos (2 [appearance: humanoid or non-humanoid] × 2 [gender: male or female]) were obtained after copyright authorization on the Internet, and two different voices were produced. Hence, eight experimental conditions were created with one news article, two news anchor appearances, two different genders, and two different voices. Figure 1 illustrates the experimental materials.

Among them, humanoid AI news anchors are jointly developed by the Xinhua News Agency and Sogou and are built on the prototype of Qiu Hao and Qu Meng, two journalists from the New Media Center. By extracting the features of lip shape and expression movement from the news broadcast video of the real anchor, Sogou used the latest AI achievements to enable virtual anchors to combine humanoid body movements for standing broadcasts with the same ability as the real-person anchor. The final presentation effect is a clone of the real anchor and an ultra-realistic replication without differences [43]. On the other hand, the non-humanoid AI news anchors were taken from the famous cartoon image of a Japanese Vtuber, which is a category of anchors who use virtual images to conduct activities on video sites. This kind of AI non-humanoid characters originated in Japan; thus, their appearances are highly animated and resemble Japanese cartoon characters [17].

For the selection and production of anthropomorphic and non-anthropomorphic voices in the experimental materials, we invited two professional Shanghai TV news broadcasters as volunteers to record the male and female versions of the news audio as the experimental materials for the two different gender voices. Then, we imported the recorded original audio into Adobe AU, a professional audio editing software. For the anthropomorphic voice, we directly exported the sound after a simple noise reduction process, while for the non-anthropomorphic voice, we selected the “Anime Sound Changing Effect” in the preset options of the software. With the assistance of a professional audio editor, we adjusted the parameters to match the cartoon characters’ voices and then exported the audio, completing four audio productions with the same quality standard. We referred to the research of Thézé [44] for the adjustment of the non-anthropomorphic voice.

In terms of the presentation of the experimental materials, all of the eight new clips kept a clarity of 1080p and similar character movements, and they only broadcasted the news simply without interaction with the audience. Besides the appearance, gender, and voice of the AI news anchors, the other contents were consistent, including video duration, volume, character size, etc.

The participants were asked to answer questions on the following three independent variables: appearance, gender, and voice of the AI news anchors. The results were analyzed by an independent sample t-test in order to confirm whether the participants could clearly perceive the differences in appearance (humanoid/non-humanoid), gender (male/female), and voice (anthropomorphic/non-anthropomorphic) of the different AI news anchors.

### 4.1. Research Procedure

Two hundred participants from a paid subject pool completed this study from 20 April 2021 to 30 April 2021(M_age_ = 27.5 years, SD = 9.5; 54.4% female). The participants were recruited online through the WenJuanXing platform (https://www.wjx.cn/, accessed on 20 April 2021), which is a professional, nationwide questionnaire platform. Due to COVID-19, we had to change the offline, face-to-face investigation to an online one.

In the manipulation check, 50 participants were recruited to do the experiment. This paper asked the following questions: “What do you think of the cartoon degree of the AI host image in this news?”, “What do you think of the anthropomorphic sound of this news broadcast?”, and “What do you think is the gender of the newscaster?”. From the three items, the first two were graded on a Likert scale of 5, while the gender items were graded using multiple-choice questions. This study also innovatively added the following: “Do you think this AI news anchor looks good? That is, how beautiful/handsome is her/his appearance?”, in order to understand the subjects’ perceived aesthetic degree of the different images. The results show the following: there are significant differences between the virtual AI news anchors and the humanoid AI news anchors (M_humanoid AI news anchors_ = 1.33, M_non-humanoid AI news anchors_ = 4.38, *n* = 50, *p* = 0.000 < 0.05); there are significant differences in the audience’s perception between the male and female AI news anchors (M_male AI news anchor_ = 1, M_female AI news anchor_ = 2, *n* = 50, *p* = 0.000 < 0.05); there are significant differences between the anthropomorphic AI news anchors and the mechanical AI news anchors (M_anthropomorphic AI news anchors_ = 4.50, M_non-anthropomorphic AI news anchor_ = 1.49, *n* = 50, *p* = 0.000 < 0.05). However, for the different appearances of the AI news anchors, there is no significant difference in the audience’s perceived aesthetic degree (M_aesthetic perception of AI news anchor image_ = 4.32, M_aesthetic Perception of Humanoid AI News Anchor Image_ = 4.31, *n* = 50, *p* = 0.863 > 0.05). To some extent, it can be ruled out that the deviation of the follow-up experimental results is caused by the “face value”. To summarize, the manipulation of the three independent variables in this study is effective.

Therefore, this study employed a multi-factor analysis of variance to test the influence of the three independent variables (appearance, gender, and voice) on the dependent variables, as well as whether any interaction occurs between them. This study used a 2 (appearance) × 2 (gender) × 2 (voice) analysis of variance (ANOVA) to determine the influence of appearance, gender, and voice on viewing intentions and to explore whether audiences perceive the appearance, gender, and voice of AI news anchors as attractive.

### 4.2. Results and Discussion

***Appearance.*** It was found that the appearance of the virtual AI news anchors significantly influences an audience’s perception of the news anchor’s attractiveness (F = 786.283, *p* < 0.01). Therefore, without considering the influence of gender and voice, the audience’s perceived attractiveness of the virtual AI news anchor image is significantly higher than the same audience’s perceived attractiveness of the humanoid AI news anchor image. That is, appearance has significant and positive effects on perceived attractiveness. Hypothesis H1 is, thus, verified.

***Gender.*** Male and female AI news anchors also had a significant impact on the audiences’ perception of attractiveness (F = 2195.696, *p* < 0.01). Therefore, without considering the influence of appearance and voice, an audience’s perception of the attractiveness of female AI news anchors is significantly greater than that of male AI news anchors. That is, gender has significant and positive impacts on the perception of attractiveness. Hypothesis H2 is, hence, verified.

***Voice.*** The anthropomorphic and non-anthropomorphic sounds also significantly influenced the audience’s perception of the attractiveness of the AI news anchors (F = 692.925, *p* < 0.01). Therefore, without considering the effects of gender and image, the audience’s perception of the attractiveness of anthropomorphic AI news anchors’ voices is significantly greater than that of mechanical AI news anchors’ voices. That is, the type of sound also has significant and positive impacts on the perception of attractiveness. Consequently, hypothesis H3 is verified. Figure 2 illustrates the estimated marginal mean of three independent variables.

The results revealed that voice–image, voice–gender, and image–gender interaction terms do not have a significant impact on an audience’s perception of the attractiveness of AI news anchors. The significance of the interaction terms of image, gender, and voice was *p* = 0.000 < 0.05, indicating that the interaction terms of the three independent variables (image, gender, and voice) have a significant impact on an audience’s perception of the attractiveness of AI news anchors. Table 1 illustrates the interaction terms among voice, gender, and image.

## 5. Study 2

Study 2 focused on the mediating and moderating effects on the viewers’ willingness to watch AI news anchors. According to previous research findings, perceived attractiveness plays an important role in message processing. It can be a strong mediator to influence people’s attitudes. Moreover, the difference in inherent impressions will affect an audience’s evaluation of things and their extensions. Since news anchors comprise the main body of AI reporting news, the inherent impression of news anchors may adjust people’s willingness to watch AI news. Therefore, this study tested the inherent impression of traditional news anchors as an alternative explanation, testing the belief that the traditional image of news anchors has an impact on the perception of AI news anchors. We tested this alternative explanation and the mediation effect through two questionnaires.

### 5.1. Research Procedure

Another 200 participants were enrolled to complete Study 2 through the WenJuanXing network (M_age_ = 29.5 years, SD = 10.6; 47.4% female). Study 2 used the same experimental materials as Study 1 (eight video clips of AI news anchors) to maintain coherency with Study 1.

We randomly distributed eight groups of online questionnaires to all of the participants. The participants were first asked to watch a piece of news broadcast by an AI news anchor edited to compare with their appearance, gender, and voice. We told the participants that the eight different AI news anchors relied on the same database and operated autonomously.

As a behavioral dependent variable, we asked the participants if they had a different inherent impression of traditional news anchors (yes or no), as well as what their inherent impression of news anchors was. We then measured the hypothesized mediator (perceived attractiveness) by asking the participants to rate the extent to which they thought that AI news anchors were visually appealing, attractive, and interesting.

At the very end of the survey, the participants were required to evaluate the AI news anchor and their willingness to watch AI news (1 = “strongly disagree” and 7 = “strongly agree”). At the end, all of the participants were asked to answer a few demographic questions.

To ensure that the reliability and validity of the questionnaire settings are in compliance with the relevant regulations on the reliability of the research and the formal experimental process is carried out smoothly, this research conducted a preliminary experiment from 30 April to 15 May 2021. Each experimental group invited five participants online to discover the questionnaire item settings and the scientificity of the experimental process. The validity of the 40 questionnaires was 100%. The data analysis of the pretests was carried out using SPSS26.0. The reliabilities of each scale Cronbach α are shown in the table below, all of which are greater than 0.9; the KMO values of the validity are shown in the table below, all of which are greater than 0.6. This means that the design of the questionnaire items is reasonable and the scales do not need to be adjusted or deleted.

The present study used the Hayes’ adjustment analysis model (PROCESS model 14), which treated perceived attractiveness as the independent variable, viewed willingness as the dependent variable, and analyzed the inherent impression of news anchors as the moderating variable. To test and verify the mediating role of perceived attractiveness, we verified the three paths of the mediating role.

### 5.2. Results and Discussion

***Mediation effect.*** The results verified the positive effects of appearance, sound quality, and gender on perceived AI attractiveness. Taking voice as the independent variable, perceived attractiveness as the mediating variable, viewing intention as the dependent variable, and the inherent impression of news anchors as the moderating variable, the mediating effect of perceived attractiveness on the influence of AI news anchors’ voices on viewing intention was verified. The fitting result R^2^ = 0.3922, *p* = 0.000 < 0.05, and the results of the lower and upper limits of the 95% confidence interval did not cross 0, which proves that perceived attractiveness plays a mediating role in the influence of AI news anchors’ voices on viewing intention. Table 2 and Table 3 illustrate the results of the mediating effect.

Taking appearance as the independent variable, perceived attractiveness as the mediating variable, viewing intention as the dependent variable, and the inherent impression of news anchors as the moderating variable, the mediating effect of perceived attractiveness on the influence of AI news anchors’ appearances on viewing intention was verified. The fitting results, R^2^ = 0.2885, *p* = 0.000 < 0.05, and the results of the lower and upper limits of the 95% confidence interval did not cross 0, which proves that perceived attractiveness plays a mediating role in the influence of the appearance of AI news anchors on viewing intention.

Gender was taken as the independent variable, perceived attractiveness as the mediating variable, viewing intention as the dependent variable, and the inherent impression of news anchors as the moderating variable, and the mediating effect of perceived attractiveness on the influence of AI news anchors’ appearances on viewing intention was verified. The fitting results, R^2^ = 0.0051, *p* = 0.3497 > 0.05, were obtained as follows: the lower and upper limits of the 95% confidence interval crossed 0, indicating that perceived attractiveness does not play a mediating role in the influence of the AI news anchor’s gender on viewing intention.

In conclusion, among the three paths of the AI news anchor’s perceived attractiveness on viewing intention, the mediating effect was only confirmed in two of the paths, with appearance and voice as the independent variables. Therefore, H6 is partially verified.

***Moderation effect.*** After 5000 instances of resampling, the independent method was used to analyze the news in order to verify the moderating effect of the audience’s inherent impression of news anchors. The fitting results were R^2^ = 0.2554, *p* = 0.0000 < 0.05, and the results are shown in Table 4.

The results revealed that perceived attractiveness had a significant, positive impact on the audience’s willingness to watch (coeff = 1.7047, t = 27.4556, *p* = 0.000 < 0.05). That is, the more attractive the audience perceives the AI news anchor to be, the greater their willingness to watch it. Hypothesis H4 is, thus, verified.

Furthermore, the audience’s inherent impression of news anchors and their perception of AI attractiveness had a significant negative impact on their willingness to watch (coeff = −0.6174, t = −36.7748, *p* = 0.000 < 0.05). That is, when an audience’s perceived attractiveness to AI is high but their inherent impression of traditional news anchors is also high, their willingness to watch AI news anchors will be significantly reduced. An audience’s inherent impression of news anchors, hence, significantly and negatively regulates the relationship between perceived attractiveness and willingness to watch. Therefore, Hypothesis H5 is confirmed.

## 6. Conclusions and Discussion

This article has presented some of the key aspects of news anchor characters, both human- and machine-like, and their perceived attractiveness to viewers. Through this process, the connection between humans and technology becomes stronger, but the impact of a human-like image still stands in its way.

### 6.1. AI News Anchors

Different from the speculative discussion of AI news anchors based on cognitive response models and stereotype theory, this study used experimental methods to study audiences’ perceived attractiveness of AI news anchors and to explore audiences’ willingness to watch those types of anchors. Based on the theoretical framework of the cognitive response model, this study innovatively introduced stereotype theory, proposed the moderating variables of audiences’ inherent impressions of news anchors, studied the influence of audiences’ different psychological, inherent cognitions of news anchors on their viewing intentions, and identified innovative strategies for AI news anchors’ current development. From the research findings, we conclude that the appearance, gender, and voice of AI news anchors, as independent variables, play significant roles in an audience’s perceived attractiveness of those news anchors. In the present study, the audience preferred virtual news anchors with a female appearance and an anthropomorphic voice, which provides practical guidance for improving the influence of AI news anchors. In the presence of two-by-two interactions between the three independent variables, the significant differences disappeared. However, the interaction terms of the three independent variables (image, gender, and voice) had a significant impact on the audience’s perception of the attractiveness of AI news anchors. In the era of AI news anchors, this paper intended to explore the attractiveness elements and the way in which each element shapes the figure of AI news anchors. Based on the data analysis, three findings are summarized in the section below.

### 6.2. Human-Like Characteristics

First, the study found that the virtual image of AI news anchors is both more attractive than human-like AI news anchors and more preferable to audiences. Virtual female AI news anchors that used anthropomorphic voices for news broadcasts gained higher audience perception and appeal. The study verifies that in the era of intelligent media, the public’s preference for traditional news anchors has transferred to the characteristics of virtual news anchors, and the comparison between virtual and humanoid appearances of anchors is one of the main innovations of this paper. Based on the analysis of the experimental results and combined with the related theory of social psychology, the study found that an audience that prefers the virtual image of higher perceived attractiveness may be associated with the generalization effect. As a classic theory of social psychology, the generalization effect refers to a certain reaction (including personal cognitive judgment, behavior, and an attitude change, among other things), which will occur after one forms a connection with a certain type of “stimulus” [45]. In the process of selecting the experimental materials, compared with humanoid AI news anchors, a virtual AI news anchor was used for the world’s first AI virtual idol, and the image of computer-synthesized virtual idols, including Luo Tianyi, who is almost “known to everyone”, has become popular in the cyberspace. Therefore, although the launching dates of a human-like AI news anchor (November 2018) and a virtual AI news anchor (November 2017) are similar, the appearance characteristics of virtual idols have been deeply rooted and accepted by most audiences. The image of a human-like AI news anchor is relatively unfamiliar to audiences. Therefore, during this experiment, when the audiences saw the image of the virtual AI news anchor, they may have been affected by the generalization effect and may have associated the news anchor with their favorite virtual idols, thus, producing a higher perceived attractiveness.

Second, with the moderating effect of the inherent cognition of traditional news anchors, there is a decrease in the audiences’ intentions to view AI news anchors. This study found that when an audience has a clear inherent impression of news anchors, even though they believe the virtual AI news anchors have a high perceptual attraction, they still do not have a strong desire to watch them. In contrast, for humanoid AI news anchors with low perceived attractiveness, the audiences have higher viewing intentions when they have positive inherent impressions of the news anchors. In response to this phenomenon, the stereotype theory is introduced. It highlights that when people have formed a relatively fixed understanding and impression of something (in this work, news anchors as a specific group), in the face of change and with the emergence of different situations, they will stick to their inherent point of view. In the selection of the materials, the appearance of the virtual AI news anchor was distinct from that of the human-like AI news anchor in terms of facial expression, overall appearance, and body language, showing the characteristics of liveliness and youthfulness. The humanoid AI news anchors were completely modeled according to the appearance of traditional, real news anchors. They had normal features with mature adult faces around 30 years old; their hairstyles were also more traditional and their clothes were formal and serious. Moreover, although they had not yet achieved the imitation of human emotions [16], their body language and facial expression control were in place and similar to those of news anchors on current traditional TV screens. The experimental data suggest that some audiences have formed an inherent impression of news anchors (i.e., that they are serious and dignified) based on experience. Therefore, when faced with the image of brand-new virtual AI news anchors, audiences reject watching them. Even though many audiences think that the image of virtual AI news anchors has high perceptual appeal, they do not choose to watch the news broadcast by such young and lively virtual idols because the idols violate the audiences’ expectations of news anchors.

Third, according to the results, the three characteristics of AI news anchors showed significant differences, but the significance of the 2–2 interaction term disappeared. This finding may be related to the halo effect when audiences form perceptions and make judgments. The halo effect refers to the fact that in the process of interpersonal communication, one will remember another’s obvious characteristics so deeply that other characteristics will be ignored, thus, producing a “partial generalization” recognition. The conditions for the formation of such illusions are often caused by insufficient mastery of information and a generalization of personal subjective inferences [46]. When faced with several AI news anchor videos at the same time, audiences are likely to build a relatively strong preference for one of the characteristics and ignore the other characteristics, thereby producing certain “Love me, love my dog” psychological effects [47]. Therefore, affected by the general psychological cognition, the characteristics of the single image in this study revealed a significant difference. However, when the two features were separate extractions for the data analysis, one feature may have been ignored by the audience due to the presence of the halo effect, leading to the non-significant results.

### 6.3. Theoretical and Practical Implications

In news theory and practice research, the question of how to make AI news anchors more attractive and have more news communication power is a hot topic. This paper aimed to identify the influencing factors of AI news anchors’ perceived attraction by audiences and to provide practical guidance for improving the influence of AI news anchors. For these anchors to broadcast news, the information about the AI’s characteristics that the audience observes and receives is diverse, including gender, appearance, facial features, hairstyle, and various other elements, such as voice, body language, and dressing. Therefore, it is not possible to construct AI news anchors simply from a single dimension. Instead, one must consider the subjective feelings and inherent psychological cognition of the audience.

Based on this, the present article explored the influencing factors behind an audience’s perceived attractiveness of AI news anchors, attempted to understand the future development trends of new AI news anchors, and discussed the changes and impacts it will bring to the news industry. The research results found that virtual female AI news anchors who use anthropomorphic voices to broadcast the news obtained the highest perceived attractiveness among the audiences. However, this image conflicts with the public’s inherent impression of news anchors, which leads to a decrease in their willingness to watch. “How to balance the conflict between perceived attractiveness and inherent impression?” has become a major question raised by this research for practitioners in the news industry. The quantitative result of this research is expected to enrich the current academic research on AI news anchors and spread the psychology of the cognitive reaction model as the basic theoretical framework. The results are also arranged to provide useful suggestions for news producers and AI researchers to build a better image and reaction model. Through the exploration of people’s inner perceptions of different AI news anchors, we hoped that this research could provide practical guidance for the re-innovation of AI news anchors in the future and deepen the research field of communication effects in the context of human–computer interactions.

### 6.4. Limitations and Proposals for Future Research

There were several limitations in this study. First, this research was conducted locally in China through an online experiment due to the COVID-19 epidemic. The research data and the number of participants were limited, which resulted in the research data not being representative of people all around the world. Therefore, there might be cultural differences when compared with other countries. To explore the relationships among the variables more deeply, future studies should adopt a more comprehensive sampling method and recruit more subjects from diversified countries. Secondly, since the questionnaire data are based on subjective answers, both the dependent and independent variables in this paper are subjective indicators, and there may, thus, be measurement errors caused by different understandings and socially expected answers. Although these variables are used as dependent variables in the experimental material checks, these measures are also subjective. Therefore, we look forward to further testing the influences of different factors based on objective indicators, such as eye trackers or other technical facilities, in the future. In addition, this paper currently explores the differences in the audiences’ perception of different AI news anchor images and voices only at a theoretical level. We are now working with technology companies and media organizations, expecting to develop a more popular AI news anchor based on the theoretical findings of this paper. Empirical studies are expected to be conducted to verify the findings of this paper and better guide the practice of the news industry.

## Figures and Tables

**Figure 1 behavsci-12-00465-f001:**
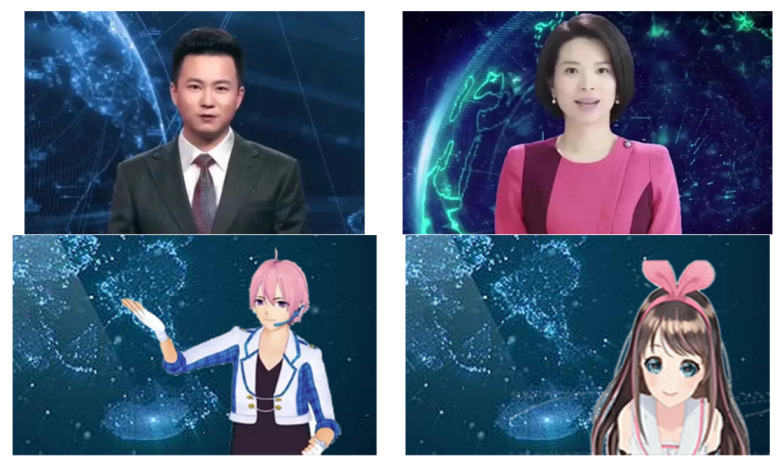
Experimental materials.

**Figure 2 behavsci-12-00465-f002:**
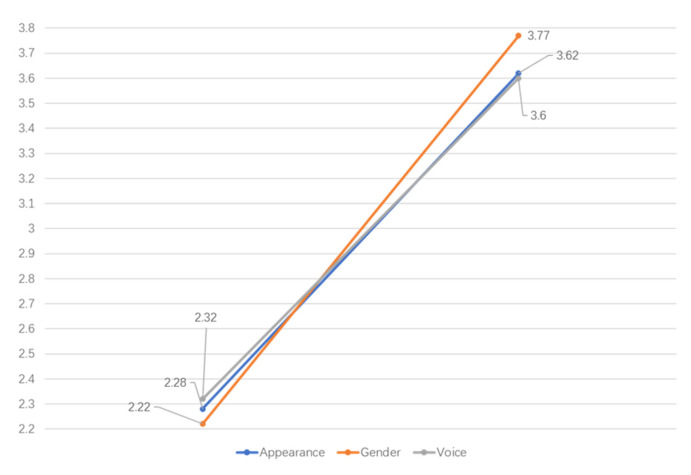
Estimated marginal mean of the variables.

**Table 1 behavsci-12-00465-t001:** Interaction terms among voice, gender, and image.

Source	Type III Sum of Squares	df (Degrees of Freedom)	Mean Squared	F	Significance
Appearance	159.782	3	53.261	786.283	0.000
Gender	148.731	1	148.731	2195.696	0.000
Voice	140.811	3	46.937	692.925	0.000
Appearance × Voice	0.456	9	0.051	0.747	0.666
Voice × Gender	0.169	3	0.056	0.831	0.478
Appearance × Gender	0.102	3	0.034	0.504	0.679
Voice × Appearance × Gender	52.140	9	5.793	85.526	0.000
R^2^ = 0.956 (Adjusted R^2^ = 0.952)					

**Table 2 behavsci-12-00465-t002:** Fit result of the mediating effect.

		Coeff	se	t	*p*	LLCI	ULCI
Perceived AI Attractiveness	Voice	0.3790	0.0320	11.8445	0.0000	0.3161	0.4419
	Appearance	0.5444	0.0425	12.805	0.0000	0.4608	0.6280
	Gender	1.2742	0.0996	12.7905	0.0000	1.0784	1.4701
	Perceived AI Attractiveness	1.7047	0.0621	27.4556	0.0000	1.5826	1.8267

**Table 3 behavsci-12-00465-t003:** Fit result of the variables in the mediating effect.

Fit Results	Effect	BootSE	BootLLCI	BootULCI
**Fit Result of Voice in Mediating Effect**	*−0.1394*	*0.0196*	*−0.1810*	*−0.1042*
**Fit Result of Appearance in Mediating Effect**	*−0.1725*	*0.0224*	*−0.2205*	*−0.1325*
**Fit Result of Gender in Mediating Effect**	*0.1227*	*0.0831*	*−0.0365*	*0.2870*

**Table 4 behavsci-12-00465-t004:** Fit results of the moderating effect.

	Coeff	se	t	*p*	LLCI	ULCI
Inherent impression of news anchors	1.8522	0.0562	32.9583	0.0000	1.7418	1.9627
Perceived attractiveness	1.7047	0.0621	27.4556	0.0000	1.5826	1.8267
Inherent impression of news anchors × Perceived attractiveness	−0.6174	0.0168	−36.7748	0.0000	−0.6504	−0.5844

## Data Availability

Not applicable.

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
