# Peer review of "What Do You Think of AI? Research on the Influence of AI News Anchor Image on Watching Intention"

_behavsci, 2022, doi:10.3390/bs12110465_

Round 1
Reviewer 1 Report
This paper treats the topic of clarifying what are the factors contributing to AI news anchors. AI news anchors are now being introduced into our life such as TV news shows. The authors carried out psychological experiments comparing various factors that influence AI news anchors. These factors include Appearance: humanoid or non-humanoid, Gender: male or female, and Voice: anthropomorphic or non-anthropomorphic. The psychological experiments are well designed, and the obtained result is simple and clear. Therefore, as a research paper from psychological aspect of view, major revision is required.
However, when looking at the paper from “common sense point of view,” there are several serious problems with this paper and the paper with the present content cannot be accepted.
First thing is that there are no figures showing the humanoid and non-humanoid. Humanoid and non-humanoid cover wide areas and it is crucial for the authors showing figures of them. What they really look like is so important and influences so much to the psychological experiments results. Probably how they actually look like is so important rather than the actual comparison between humanoid and non-humanoid. Also, the detailed description of them are missing. Does humanoid means recorded human , or human-like robot, or CG character? Do they have interactive capability or not?
Also, for male and female, the authors should provide their appearances. Are they computer characters? Is there good correspondence between their CG quality and their behaviors. Otherwise, it is difficult for the readers to judge whether the obtained results are based on the difference between genders, or the qualities.
And also, for voice, what are anthropomorphic and non-anthropomorphic voices? Does anthropomorphic voice mean actual human voice? Recently as voice synthesis technology has advanced a lot, the authors should have used synthesized voice, if they wanted to achieve a fair comparison between anthropomorphic and non-anthropomorphic voices. It looks like the authors used voice changer software to create non-anthropomorphic voices. Please clarify the name of the software. It is apparent that voice changer decreases the quality of voice. Therefore, it is natural that the subjects evaluated non-anthropomorphic voices lower than anthropomorphic voices.
Again the psychological experiments themselves are well-designed and well achieved. However, the stimulus used in the experiments are unclearly described, and therefore it is not clear whether this paper can contribute to the area of AI and human behaviors.
Author Response
Dear Reviewer,
Thank you for your comments on our manuscript entitled “What Do You Think of AI?Research on the Influence of AI News Anchor Image on Watching Intention” (ID: behavsci-1992303). These comments are very valuable for modifying and perfecting our paper, and also contribute to the important guiding significance of our research. We have carefully studied the comments and made corrections, hoping to meet with approval. The revised part has been marked up using the “Track Changes” in the paper. The main corrections in this article and the responses to your comments are as follows:
Point 1: There are no figures showing the humanoid and non-humanoid. Humanoid and non-humanoid cover wide areas and it is crucial for the authors showing figures of them. What they really look like is so important and influences so much to the psychological experiments results. Probably how they actually look like is so important rather than the actual comparison between humanoid and non-humanoid. Also, the detailed description of them are missing. Does humanoid means recorded human , or human-like robot, or CG character? Do they have interactive capability or not?
Response 1: It is really true as you suggested that showing experimental materials is very important for the results of psychological experiments. Therefore, we added all the experimental materials to the "4. Study 1" section, including male humanoid image, female humanoid image, male non-humanoid image, and female non-humanoid image. Since we can't show the video and audio in the paper, we further described the detailed information of the experimental materials and the significant differences among different experimental materials by means of text introduction. Please see the descriptions in the paper for more details about the appearance. For example, the source of the images of the humanoid AI news anchor, whether they are CG animation, and whether they have interactive capability, etc.
For the points about "How they actually look like is so important rather than the actual comparison between humanoid and non-humanoid. " mentioned in the review comments. In the second paragraph of "4.1 Research Procedure", the "manipulation check" section, we used question "Do you think this AI news anchor looks good? That is, how beautiful/handsome is her/his appearance?", to understand the subjects' perceived aesthetic degree of different images. The results of the pre-experiment showed that, for different appearances of AI news anchors, there are no significant difference in the audience's perceived aesthetic degree (Maesthetic perception of AI news anchor image =4.32, Maesthetic Perception of Humanoid AI News Anchor Image =4.31, n=50, p=0.863>0.05). To some extent, it can be ruled out that the deviation of follow-up experimental results is caused by difference in appearance rather than being humanoid or non-humanoid.
Point 2: For male and female, the authors should provide their appearances. Are they computer characters? Is there good correspondence between their CG quality and their behaviors. Otherwise, it is difficult for the readers to judge whether the obtained results are based on the difference between genders, or the qualities.
Response 2: As reviewer suggested that, we have put in figures of four different experimental materials, while the picture quality of the video and the interactivity of the experimental materials have been described additionally.
Also, manipulation check was carried out and was reported in "4.1 Research Procedure" the extent to which different experimental condition groups can distinguish between appearance (humanoid vs. non-humanoid), gender (male vs. female) and voice (anthropomorphic vs. non-anthropomorphic).
Point 3: For voice, what are anthropomorphic and non-anthropomorphic voices? Does anthropomorphic voice mean actual human voice? Recently as voice synthesis technology has advanced a lot, the authors should have used synthesized voice, if they wanted to achieve a fair comparison between anthropomorphic and non-anthropomorphic voices. It looks like the authors used voice changer software to create non-anthropomorphic voices. Please clarify the name of the software. It is apparent that voice changer decreases the quality of voice. Therefore, it is natural that the subjects evaluated non-anthropomorphic voices lower than anthropomorphic voices.
Response 3: As reviewer suggested that, we have described in detail the different ways of acquiring anthropomorphic and non-anthropomorphic voices and the specific production methods, including the names of the processing software.
For the difference in sound quality, the audio recording conditions as well as the audio production software chosen for this paper were supported by the hardware of our research facility, a professional recording studio and professional-grade film sound production software. At the same time, this paper also tested the video quality and audio quality in the manipulation test, and the results showed that there was no significant difference in audience perception. Therefore, errors in experimental results due to differences in video quality or sound quality were excluded.
Special thanks to you for your comments.
We strive to improve the manuscript based on the suggestions. These changes will not affect the content and framework of the whole paper. Here we did not list the detailed changes but marked up using the “Track Changes” in the paper. We sincerely appreciate your warm work earnestly and hope that the corrections will meet with approval. Thanks again for your comments and suggestions.
With best regards,
Yifei Li
School of Media and Communication, Shanghai Jiao Tong University
800 Dongchuan Rd., Shanghai 200240, P. R. China.
e-mail: [email protected]

Reviewer 2 Report
The article addresses an extremely relevant topic in the contemporary field of media and technology. However, a necessary ethical discussion about the adoption of these technologies in the field of journalism and information, which is indispensable to a human-centered technological development, is absent. In this sense, the article does not question why better image and reaction models are necessary and what effective benefits they bring, showing an evident tendency of technological determinism. The introduction section exposes numerous premises that need explicit substantiation in the light of international bibliography. At the methodological level, it is fundamental to expose the limits associated with the use of the WenJuanXing Network as a basis for recruiting participants.
Author Response
Dear Reviewer,
Thank you for your comments on our manuscript entitled “What Do You Think of AI?Research on the Influence of AI News Anchor Image on Watching Intention” (ID: behavsci-1992303). These comments are very valuable for modifying and perfecting our paper, and also contribute to the important guiding significance of our research. We have carefully studied the comments and made corrections, hoping to meet with approval. The revised part has been marked up using the “Track Changes” in the paper. The main corrections in this article and the responses to your comments are as follows:
Point 1: A necessary ethical discussion about the adoption of these technologies in the field of journalism and information, which is indispensable to a human-centered technological development, is absent.
Response 1: It is really true that the ethical discussion about the adoption of AI technology in the field of journalism is necessary for our research. Therefore, we have added relevant discussions and classical literature in this academic field in the second paragraph of "1. introduction" section.
Point 2: The article does not question why better image and reaction models are necessary and what effective benefits they bring, showing an evident tendency of technological determinism.
Response 2: As you suggested that, we have provided a detailed discussion of the current research gaps in the field of AI news anchors, the gaps between industry practice and public expectations, and the research value that this paper is expected to bring. For more details, please see the third and fourth paragraphs of "1. introduction" section and "5.3. Theoretical and practical implications" section.
Point 3: The introduction section exposes numerous premises that need explicit substantiation in the light of international bibliography.
Response 3: Thank you for your suggestions on the supplement to the international literature. Since China is currently the most widely used country for AI news anchors, we found that more research on AI news anchors was being conducted by Chinese scholars, resulting in a slightly lower percentage of international bibliography. In any case, we conducted a detailed search on AI news anchors once again in the international academic community and supplemented some cutting-edge academic research in the "1. introduction" section.
Point 4: At the methodological level, it is fundamental to expose the limits associated with the use of the WenJuanXing Network as a basis for recruiting participants.
Response 4: We are very sorry for our negligence of the detailed description of research limitations, and we have discussed our research limitations in detail and proposed future research plans accordingly in the " 5.4. Limitations and proposals for future research " section
Special thanks to you for your comments.
We strive to improve the manuscript based on the suggestions. These changes will not affect the content and framework of the whole paper. Here we did not list the detailed changes but marked up using the “Track Changes” in the paper. We sincerely appreciate your warm work earnestly and hope that the corrections will meet with approval. Thanks again for your comments and suggestions.
With best regards,
Yifei Li
School of Media and Communication, Shanghai Jiao Tong University
800 Dongchuan Rd., Shanghai 200240, P. R. China.
e-mail: [email protected]

Round 2
Reviewer 1 Report
Most of the reviewer's comments for the original paper have been well considered and reflected in the revised paper. Therefore the paper can be accepted in its present form.
Author Response
Dear Reviewer,
Thanks again for your comments on our research! These suggestions are very valuable for modifying and perfecting our paper, and also contribute to the important guiding significance of our research!
With best regards,
Yifei Li
School of Media and Communication, Shanghai Jiao Tong University
800 Dongchuan Rd., Shanghai 200240, P. R. China.
e-mail: [email protected]